Neural representation of face familiarity in an awake chimpanzee

Fukushima Hirokata 1 2 fukush@kansai-u.ac.jp
Hirata Satoshi 3 4
Matsuda Goh 1 5
Ueno Ari 1 6
Fuwa Kohki 3 7
Sugama Keiko 3
Kusunoki Kiyo 3 7
Hiraki Kazuo 1
Tomonaga Masaki 8
Hasegawa Toshikazu 1
1 Graduate School of Arts and Sciences, The University of Tokyo , Japan
2 Faculty of Sociology, Kansai University , Japan
3 Great Ape Research Institute of Hayashibara Biochemical Laboratories, Inc. , Japan
4 Wildlife Research Center, Kyoto University , Japan
5 JST, CREST , Japan
6 Department of Human Relations Studies, School of Human Cultures, The University of Shiga Prefecture , Japan
7 EarthMate-ChimpanzeeNEXT , Japan
8 Section of Language and Intelligence, Primate Research Institute, Kyoto University , Japan
Fadiga Luciano
Electronic publication date: 2013 Dec 10
Publication date: 2013
Volume: 1
Electronic Location ID: e223
Received 2013 Oct 3; Accepted 2013 Nov 20
Copyright: © 2013 Fukushima et al.
Copyright year: 2013
Copyright holder: Fukushima et al.
License: This is an open access article distributed under the terms of the Creative Commons Attribution License, which permits unrestricted use, distribution, and reproduction in any medium, provided the original author and source are credited.
License URL: https://creativecommons.org/licenses/by/3.0/

Keywords: Chimpanzee, Face recognition, Familiarity, Self recognition, Species effect, Memory, Comparative neuroscience, Social cognition, Event-related potentials

Funding: Grants-in-Aid for Scientific Research grant numbers 23830107 24119005 22240026 This study was financially supported by Grants-in-Aid for Scientific Research (grant numbers 23830107 and 24119005 to HF, 22240026 to GM and KH) from the Japan Society for the Promotion of Science, and the Center for Evolutionary Cognitive Science at The University of Tokyo. The funders had no role in study design, data collection and analysis, decision to publish, or preparation of the manuscript.

==============================
Evaluating the familiarity of faces is critical for social animals as it is the basis of individual recognition. In the present study, we examined how face familiarity is reflected in neural activities in our closest living relative, the chimpanzee. Skin-surface event-related brain potentials (ERPs) were measured while a fully awake chimpanzee observed photographs of familiar and unfamiliar chimpanzee faces (Experiment 1) and human faces (Experiment 2). The ERPs evoked by chimpanzee faces differentiated unfamiliar individuals from familiar ones around midline areas centered on vertex sites at approximately 200 ms after the stimulus onset. In addition, the ERP response to the image of the subject’s own face did not significantly diverge from those evoked by familiar chimpanzees, suggesting that the subject’s brain at a minimum remembered the image of her own face. The ERPs evoked by human faces were not influenced by the familiarity of target individuals. These results indicate that chimpanzee neural representations are more sensitive to the familiarity of conspecific than allospecific faces.

Introduction

As social animals, primates rely on face recognition to identify individuals and recognize their emotional states. Until recently, vast knowledge of the primate visual system has been primarily obtained from the brains of macaque monkeys and humans. Previous research has suggested that the ventrolateral visual areas are central for face recognition both for monkeys and humans (Perrett, Rolls & Caan, 1982; Tsao et al., 2003). However, the neural mechanisms responsible for face recognition in our closest living species, the chimpanzee, remain relatively unknown. Examining these mechanisms would shed new light on the physiological and evolutionary origins of human social cognition. Behavioral studies of chimpanzees have revealed that face stimuli attract their visual attention, just as in humans, indicating the importance of face perception for chimpanzees (Kano & Tomonaga, 2009). It has been further reported that chimpanzees process face configurations in a similar manner to humans (Hirata et al., 2010; Parr, 2011; Tomonaga, 2007), they understand conspecific facial expressions (Parr, 2003), and they have sophisticated abilities to identify individuals in terms of kinship detection (Parr & de Waal, 1999).

Direct examinations of the chimpanzees’ neural activities have also been conducted since the 1960s (Adey, Kado & Rhodes, 1963; Boysen & Berntson, 1985), and are currently a growing research topic (Fukushima et al., 2010; Hirata et al., 2013; Parr et al., 2009; Rilling et al., 2007; Taglialatela et al., 2008; Taglialatela et al., 2009; Ueno et al., 2008; Ueno et al., 2010). In particular, several research groups have reported neural examinations of chimpanzee visual processing (Boysen & Berntson, 1985; Fukushima et al., 2010; Parr et al., 2009). For instance, Parr and colleagues (Parr et al., 2009) conducted a functional neuroimaging study of chimpanzee face processing using positron emission tomography (PET). In their study, the subject chimpanzees were given an isotope marker and performed a matching task with conspecific faces and objects, and they were then sedated before their brains were scanned. Their results explicitly revealed that the chimpanzees recruit similar neurocognitive substrates to humans for face perception. However, measurements of neural metabolism using methods such as PET imaging are poor metrics of the temporal aspects of the neural processing. Furthermore, non-invasive measurements of primates’ brains often require sedation (Boysen & Berntson, 1985; Parr et al., 2009; Rilling et al., 2007; Taglialatela et al., 2008; Taglialatela et al., 2009), and this severely limits real-time measurement of their cognitive functions, particularly for elucidating socio-emotional responses.

We have developed a protocol of non-invasive measurement of electrophysiological activities in an adult female chimpanzee by means of scalp event-related potentials (ERPs) while she is fully awake (Fukushima et al., 2010; Hirata et al., 2013; Ueno et al., 2008; Ueno et al., 2010). Using this protocol, we previously revealed neural evidence that this chimpanzee discriminates human faces from other categories of objects, as there was a divergence of ERPs in response to face stimuli at approximately a 100 ms latency at posterior temporal regions (Fukushima et al., 2010). Furthermore, we demonstrated that chimpanzee neural activities are influenced by observing emotional expressions of conspecifics with an approximately 210 ms latency (Hirata et al., 2013).

Along this line of research, one issue to be further clarified is the influence of experience on face processing. Particularly, familiarity is considered to be the basis of individual recognition and an important factor in the modulation of social behaviors, by means of in-group and out-group differentiation, for example. Human ERP studies show that a negative deflection around 140–200 latency (‘N170’) reflects a primal stage of face processing (Bentin et al., 1996; George et al., 1996). Importantly, it has been shown that the effects of facial familiarity appear in latencies later than 200 ms after stimulus onset. That is, they appear later than the initial processing of basic facial configuration (Bentin & Deouell, 2000; Eimer, 2000; Henson et al., 2003; Schweinberger, Pfütze & Sommer, 1995). Neuroimaging studies have further elucidated that facial familiarity modulates not only ventral visual areas, but also distributed networks, including affective and memory-related regions, including the amygdala and medial parietal areas (Gobbini & Haxby, 2007; Natu & O’Toole, 2011). Concerning chimpanzees, behavioral studies have revealed that familiarity with the target enhances identification of individuals (Parr, Siebert & Taubert, 2011), but neural responses to familiarity during face perception remain unknown.

Based on these studies, the primary aim of the present study was to examine the effects of familiarity on neural activities during face processing. The first experiment in this study compared ERPs during the perception of familiar and unfamiliar conspecific faces. As a topic somewhat related to familiarity perception, self-face recognition in chimpanzees is a research topic that has received a great deal of attention (Bard et al., 2006; Gallup, 1970; Povinelli, 1987). The subject chimpanzee in the present study showed evidence of mirror self-recognition, as indicated by self-exploratory behaviors of her own body while watching a mirror (S Hirata, unpublished data). Although the present study primarily examined the effects of familiarity on face processing, the experiment also included an image of the subject’s own face as a stimulus category, as a first preliminary investigation of neural correlates of self-perception in the chimpanzee.

It is also known that the species shown in a target stimulus can influence visual processing of the displayed faces. However, whether or not primates have a specialized mechanism for perceiving the faces of conspecifics is controversial (Martin-Malivel & Okada, 2007; Parr, Heintz & Pradhan, 2008). Among previous behavioral studies in chimpanzees, some findings have suggested differential perception between conspecific and allospecific faces (Dahl et al., 2009; Fagot & Tomonaga, 1999; Fujita, 1990; Hattori, Kano & Tomonaga, 2010; Parr, Dove & Hopkins, 1998; Parr, Heintz & Akamagwuna, 2006; Parr, Siebert & Taubert, 2011; Tomonaga, Itakura & Matsuzawa, 1993), but some research has also suggested that chimpanzees process human faces in a similar manner to conspecific faces (Kano & Tomonaga, 2009; Martin-Malivel & Okada, 2007; Tomonaga, 2007). These inconsistent findings from behavioral studies motivate further elucidation of the species dependency of face processing by identifying the neural responses to conspecific and non-conspecific faces and examining possible differences in their patterns. Therefore, in the second experiment in this study, we presented the subject with faces of allospecific humans, and compared the ERP responses with familiar and unfamiliar human faces, to examine whether chimpanzee brain activity is differentially responsive to the familiarity of conspecific and allospecific faces.

Materials and Methods

Subject

The subject was a female chimpanzee (Pan troglodytes) named Mizuki, who was housed at the Great Ape Research Institute of Hayashibara Biochemical Laboratories, Inc., Okayama, Japan. She was raised by human caregivers from a few days after birth. Since the subject arrived at the Great Ape Research Institute when she was 2 years and 1 month old, she has spent the majority of her time with other chimpanzees in outdoor and indoor compounds. At the time of experimentation, the subject was 10–11 years old and living in the institute with other group members (two were male, two were female, and one was an infant). At that time, the subject had undergone other behavioral cognitive experiments (Hirata & Fuwa, 2007; Idani & Hirata, 2006) and earlier ERP experiments (Fukushima et al., 2010; Hirata et al., 2013; Ueno et al., 2008; Ueno et al., 2010). She had also participated in informal tests of mirror self-recognition, recognition of self-images on TV monitors, and match-to-sample tasks of the faces of conspecifics (S Hirata, unpublished data). A playground for the chimpanzees in the institute had a small wall covered with stainless-iron, and it was observed that chimpanzees including the present subject looked at the reflective wall and made some gestures, such as opening the mouth and looking into it.

This research was conducted in accordance with the “Guide for the Care and Use of Laboratory Animals” of Hayashibara Biochemical Laboratories, Inc., and the Weatherall report, “The use of non-human primates in research”. The research protocol was approved by the Animal Welfare and Animal Care Committee of The University of Tokyo and Hayashibara Biochemical Laboratories, Inc. (GARI-051101).

Apparatus and stimuli

The overall methodology basically followed our previous study (Fukushima et al., 2010). The experimental room consisted of concrete walls, with moderate lighting. The subject sat on a concrete platform and was fully awake during recordings. A 17-inch CRT display (IIyama LA702U) was set up in front of her approximately 40 cm away, at the horizontal level of her head. An infrared video camera was fixed on top of the CRT display to monitor the subject from a frontal view. We used this camera to check if the subject’s gaze was directed to the stimulus display.

Experimental stimuli in Experiment 1 were color photographs of the faces of chimpanzees (Fig. 1). The stimuli consisted of seven chimpanzee faces: three of unfamiliar (out-group) chimpanzees that were novel to the subject, three of familiar (in-group) chimps, and the one of the subject’s own face. The images were digitally processed in 24-bit color by graphics software. The stimuli in Experiment 2 consisted of six human faces: three of familiar persons who were the caregivers of the subject, and three of unfamiliar humans who were novel to the subject. All of the human models were Japanese subjects in their 30s, and two male subjects and one female subject were included in both categories. In both experiments, all stimuli were displayed on a black background with a visual field approximately 15° high and 13° wide. The average luminance (mean, 119.29 cd/m2; SD, 0.73 cd/m2) and number of pixels (mean, 94213; SD, 265) among stimulus categories were matched, varying less than 0.8%.

Figure 1 Stimulus images used in the present study.

In Experiment 1, the subject was presented seven images of chimpanzee faces in the three categories of unfamiliar, familiar, and selfimages. In Experiment 2, the subject was presented six images of human faces with unfamiliar and familiar categories. Note that only one example of each category is shown for Experiment 2 for privacy protection.

Procedure

Experiment 1 was conducted in three recording sessions in separate days, each consisting of five blocks with 105 trials (1575 trials in total). Experiment 2 was also conducted in three recording sessions with different days from Experiment 1, each consisting of five blocks with 60 trials (900 trials in total). On each trial of both experiments, one of the stimuli (seven images in Experiment 1 and six images in Experiment 2) was presented for 500 ms in a pseudo-randomized order with no consecutive repetition of the same stimulus. Each stimulus was followed by a 700 ms interstimulus interval (an empty black screen). To maintain the subject’s attention to the display, occasional images of several objects that were not related to faces or animals (e.g., geometric figures or patterns) were also presented randomly at a rate of once every several trials. Between blocks, the subject was given a rest period of approximately 1 min, which allowed her to make considerable body movements and to receive fruit rewards. During the recordings, an experimenter (one of the subject’s caregivers) stood beside her to keep her still and facing the display. The subject’s gaze appeared to occasionally avert from the monitor. When this occurred, another experimenter, who was monitoring the subject’s gaze direction, manually added a marker in the electroencephalography (EEG) data via a keyboard connected to the measurement computer.

ERP recording and analysis

EEG was recorded from Ag/AgCl electrodes attached to five scalp positions (Fz, Cz, Pz, T5, and T6), according to the international 10–20 system for humans. The signals were referenced to the forehead midline (FPz) and a ground electrode was positioned at the left earlobe (Fukushima et al., 2010). The electrodes were filled with Quick GEL and impedances were kept below 6 kΩ. Signals were amplified by NuAmp-40 and processed by Acquire 4.3 software (NeuroScan Inc.) with a 1,000 Hz sampling rate. A 0.1–20 Hz band-pass filter (24 dB/oct) was applied in the offline analysis. All data were segmented into 700-ms epochs, including a 100-ms pre-stimulus baseline period, based on time markers of the stimulus onset. These epochs were baseline corrected with respect to the mean amplitude over the 100-ms pre-stimulus period. Epochs that exceeded ±60 µV were excluded from the analysis. Epochs that contained ‘non-looking’ markers described above were also excluded in the analysis. The numbers of epochs accepted for the analysis were as follows: in Experiment 1, 115, 126, and 52, for unfamiliar, familiar, and self faces, respectively; in Experiment 2, 121 and 105 for unfamiliar and familiar faces, respectively.

In Experiment 1, a successive ANOVA with a single factor of stimulus type (unfamiliar/familiar/self) was applied along each data point of each channel to test differentiation among the ERP amplitudes for stimulus categories. In Experiment 2, a successive two-tailed t-test was applied to explore the difference between ERPs to unfamiliar and familiar human faces. To avoid the detection of spurious differentiation among categories, we considered a time range of 30 consecutive time points (30 ms) of p-values <0.05 to indicate a significant main effect.

Results

Experiment 1

The images of seven chimpanzee faces (three unfamiliar [out-group] chimpanzees, three familiar [in-group] chimps, and the subject herself; Fig. 1) were pseudo-randomly presented during EEG measurements. ERPs time-locked to the presentation of the face stimuli were calculated for each category (i.e., unfamiliar/familiar/self).

The basic morphologies of the ERP waveforms remarkably resembled those reported in our previous studies with the same subject (Fukushima et al., 2010; Hirata et al., 2013). The ERPs from midline sites (Fz, Cz, and Pz) showed three dominant deflections (Fig. 2): early negative components within a 100 ms latency, positive deflections peaking around a 140–150 ms latency, and a negative slow wave, which was visible after approximately a 200 ms latency. At the lateral occipitotemporal sites (T5 and T6), additional negative components appeared over the time range of 100–150 ms.

Figure 2 Averaged ERP waveforms elicited by images of unfamiliar and familiar conspecific faces as well as the subject’s own face in Experiment 1.

The blue-shaded squares overlaid on the waveforms show the periods of statistically significant main effects of stimulus category. The solid lines below the waveforms show the periods of significant difference between unfamiliar and familiar (green) and unfamiliar and self (orange) images.

A successive analysis of variance (ANOVA) with a factor of stimulus type (unfamiliar/familiar/self) was applied along each ERP data point of each channel to test differentiation among the stimulus categories. A significant main effect of stimulus type was detected over the midline electrode sites (Fz, Cz, and Pz), as indicated in Fig. 2 (all F-values >3.027 [df1 = 2, df2 = 290], p-values <0.05). This differentiation was found in the time range from approximately a 200 to 450 ms latency, most prominently at the Cz site. The details of the latencies that differentiated each category at each electrode position are depicted in Table 1. Post-hoc two-tailed t-test revealed that the waveforms for unfamiliar faces significantly diverged from those of familiar and self faces and no differences in the ERPs related to familiar and self faces were detected (Table 1).

Table 1 Time ranges where ERPs for each category differed significantly in Experiment 1 (post stimulus latencies in ms).

Test	Electrodes	Time ranges of statistical
significance	
3-levels ANOVA	Fz	232–269, 286–309	
	Cz	201–317, 393–445	
	Pz	232–269, 402–438	
Unfamiliar vs. Familiar	Fz	232–310	
	Cz	192–309	
Unfamiliar vs. Self	Cz	201–325, 392–453	
	Pz	396–453	
	T6	393–434	

Experiment 2

Following Experiment 1, in which we examined the effects of familiarity in ERP responses to conspecific faces, Experiment 2 was conducted to investigate the same effects of familiarity in ERP responses to allospecific human stimuli. The images of six human faces (three persons familiar and three persons unfamiliar to the subject) were pseudo-randomly presented using the same basic design as in Experiment 1.

Figure 3 illustrates ERP waveforms elicited by the images of familiar and unfamiliar human faces. A successive t-test (df = 224) along each data point of the ERP waveforms on each channel failed to detect any significant differences in the ERPs related to familiar and unfamiliar human faces.

Figure 3 Averaged ERP waveforms elicited by images of unfamiliar and familiar human faces in Experiment 2.

Discussion

This study examined whether and how the familiarity of faces is reflected in chimpanzee ERPs. Based on the results of Experiment 1, we propose that familiarity modulates the ERPs evoked by conspecific faces, differentiating faces of familiar subjects from the subject’s own face. The familiarity effect was observed in a time range later than approximately 200 ms post-stimulus onset. Our previous study reported that face-specific modulation of the chimpanzee’s visual ERP was initiated as early as around a 100 ms latency in comparison with images of scrambled faces or non-face objects (Fukushima et al., 2010). Thus, the results of our previous study and the current study together indicate an early response to face processing and relatively late modulation by face familiarity. This pattern is consistent with human studies, in which ERPs differentiate familiar and unfamiliar faces in a time period after the primary processing of face configuration (i.e., after the occurrence of the N170 component) (Bentin & Deouell, 2000; Eimer, 2000; Henson et al., 2003; Kaufmann, Schweinberger & Burton, 2009; Miyakoshi et al., 2008; Paller et al., 2003), although a few research groups have reported earlier latencies of this familiarity effect (Caharel et al., 2002; Caharel et al., 2005; Jemel, Schuller & Goffaux, 2010; Leleu et al., 2010). It is considered that the familiarity effect basically occurs in later stages of face recognition because it interacts with multiple face-related processing steps (Gobbini & Haxby, 2007). Interestingly, the latency of the familiarity effect in the present study resembled that of our previous study of affective processing, in which the ERP differentiation of affective vs. non-affective pictures initiated about a 210 ms latency (Hirata et al., 2013). Thus, the familiarity effect in the current study may be partially modulated by affective processing.

With respect to its topography, the familiarity effect was observed around the midline electrode sites (i.e., Fz, Cz, and Pz) centered on the vertex. This result was different from our previous study (Fukushima et al., 2010), which demonstrated face-specific patterns around temporal and posterior regions (i.e., T5/T6 as well as Pz electrode sites), suggesting that the familiarity effect in the current study was modulated by neural substrates not limited to those specific to face processing. This consideration is again consistent with neuroimaging studies in human subjects, which indicate that the familiarity effect is subserved by non-visual networks, such as memory or affective networks (Gobbini & Haxby, 2007; Natu & O’Toole, 2011). Another related study from our group examined the ERPs of the same subject while they received auditory presentations of the names of familiar (including her own name) and unfamiliar chimpanzees, showing that the evoked potentials from fronto-central sites differentiated unfamiliar names from familiar ones (Ueno et al., 2010). Although the previous auditory study used different reference-channels (i.e., A1/A2) from the current visual study, it is possible that similar mechanisms subserved familiarity processing across the two studies. Furthermore, a few human ERP studies of face perception have reported a familiarity effect not only at temporo-occipital sites but also in regions around fronto-central sites (Caharel et al., 2002; Miyakoshi et al., 2008). It is difficult to strictly compare ERP topographies from chimpanzees and humans because of structural differences in terms of the shapes of the bone and muscles of the skull between the species (Burrows et al., 2006), and these differences may substantially influence scalp-surface potentials. However, the results of the present experiments are somewhat consistent with those of previous human studies.

It is noteworthy that the ERPs elicited by the subject’s own face diverged from those evoked by unfamiliar but not familiar faces of conspecifics in Experiment 1. Human studies have shown that self-face perception is reflected in the modulation of face-evoked ERPs, although the latency and topography of this effect are somewhat inconsistent among reports (Caharel et al., 2002; Keyes et al., 2010; Ninomiya et al., 1998; Sui, Zhu & Han, 2006). In our previous study mentioned above, the ERPs from the same subject chimpanzee showed specific patterns in their response to the auditory presentation of the subject’s own name with a 500 ms latency, suggesting that the chimpanzee brain at least differentiated this self-relevant stimulus (Ueno et al., 2010). The reason why the current study failed to detect self-specific patterns in terms of visual face recognition may be quantitative differences in the amount of exposure to this stimulus. Auditory symbols had been much more frequently used in the subject’s communication with human caregivers. As for the visual recognition of self-image, the subject chimpanzee was exposed to the image of its own face in a few experimental sessions (S Hirata, unpublished), and she also had occasional access to visual images of her own face (see Methods). However, the visual input of her own face may not be sufficient to differentiate ERPs related to self and familiar non-self faces. This study at least suggests that experience with self-face perception can influence neural representations of the self-face by shifting it toward that of “familiar” individuals. In other words, we cannot claim based on our results that the subject chimpanzee recognized an image of her own face as representing herself, but we can claim that the subject’s brain did respond differently to her own face. Determining whether there is a chimpanzee neural signature of self-face recognition requires further investigation using matching experience with the stimuli, and examination of the correspondence between neural activities and measures of self-oriented behaviors.

In contrast to the responses to the chimpanzee faces, ERPs to human faces did not differentiate familiar and unfamiliar individuals (Fig. 2). This result suggests that modulation of face familiarity in the chimpanzee visual system depends on the species displayed in the target stimulus; that is, the chimpanzee was more sensitive to familiarity of conspecific faces than allospecific faces. Behavioral studies in non-human primates have shown that monkeys have a relatively robust preference for conspecific faces despite substantial exposure to other species, even when they lack visual experiences of conspecific faces (Fujita, 1993; Martin-Malivel & Okada, 2007; Sackett, 1966). Chimpanzees have also been shown to have self-species superiority in social cognition, in terms of, for example, fast face recognition (Parr, Dove & Hopkins, 1998; Tomonaga, Itakura & Matsuzawa, 1993) and attracting initial looking at a conspecific face (Hattori, Kano & Tomonaga, 2010). Therefore, the differences between chimpanzee vs. human faces in the current study are in line with those reports, and can be further assumed to suggest that visual face processing is an inherent property of chimpanzees. However, in parallel to this view, a few studies have suggested the presence of flexible modulation of cross-species face perception in chimpanzees. For example, captive chimpanzees show similar configural processing to human faces as to conspecific faces (Parr, Dove & Hopkins, 1998), and chimpanzees reared by humans show better discrimination of human faces than chimpanzee faces (Martin-Malivel & Okada, 2007). Furthermore, recent studies have explicitly examined the effects of experience on cross-species face processing, showing that young chimpanzees predominantly process conspecific faces, whereas old chimpanzees predominantly process human faces (Dahl et al., 2013a; Dahl et al., 2013b). Their mathematical simulations predicted that this shift in face processing occurs around the age of 15 years in chimpanzees (Dahl et al., 2013a). The subject of our study was 10–11 years old at the time of the experiments, thus it is reasonable to speculate that she still had a dominant sensitivity to conspecific faces, and this was reflected in the ERP responses. Such flexibility is also observed in humans, as it has been shown that human infants have a bias to conspecific faces (De Haan, Pascalis & Johnson, 2002), but it may be also flexible as human adults perceive chimpanzee faces with a quite similar process to that used for human faces (Carmel & Bentin, 2002). Further investigations of face processing mechanism of primates should take into account interactions between innate tendencies and acquired properties (De Haan, Humphreys & Johnson, 2002).

Finally, we should be cautious of a couple of issues when interpreting the present findings. First, it may be the case that neural differentiation of human familiarity did occur in Experiment 2 but it was so small that it was below the limit of detection of EEG or occurred in a manner that is not amenable to measurement by cross-trial averaging of scalp-surface potentials (Luck, 2005; Tallon-Baudry & Bertrand, 1999). Therefore, it is still premature to conclude that the chimpanzee is differentially sensitive to the conspecific and allospecific faces. Second, as pointed out in our earlier discussion of the response to the self-face image, it is also possible that our results simply reflect quantitative differences in the amount of exposure to the target categories. The present subject was reared by humans, and interacted every day with the human staff whose faces were presented as the stimuli of the “familiar” category in Experiment 2. Thus, we suggest that the subject was very familiar with these human faces, and it is valid to define the set of pictures depicting her caregivers as the “familiar” category for this subject. However, the amount of exposure to the familiar chimpanzees and humans, particularly the total amount of time previously spent looking at their faces, could not be exactly and quantitatively matched in this study. In common with the topic of self-face recognition, the current study was a preliminary attempt to clarify the effects of familiarity on face processing and cross-species differences in these effects. Further examinations of how experience modulates face perception should apply strict control over exposure to the target stimuli and behavioral indices.

Additional Information and Declarations

Competing Interests

Author Contributions

Animal Ethics

Kohki Fuwa and Kiyo Kusunoki run (or conduct) EarthMate-ChimpanzeeNEXT. Satoshi Hirata, Kohki Fuwa, Keiko Sugama, Kiyo Kusunoki were employees of Great Ape Research Institute of Hayashibara Biochemical Laboratories, Inc.

Hirokata Fukushima and Goh Matsuda conceived and designed the experiments, analyzed the data, wrote the paper.

Satoshi Hirata conceived and designed the experiments, performed the experiments, wrote the paper.

Ari Ueno conceived and designed the experiments.

Kohki Fuwa, Keiko Sugama and Kiyo Kusunoki performed the experiments.

Kazuo Hiraki, Masaki Tomonaga and Toshikazu Hasegawa supervised the project.

The following information was supplied relating to ethical approvals (i.e., approving body and any reference numbers):

The research protocol was approved by the Animal Welfare and Animal Care Committee of The University of Tokyo and Hayashibara Biochemical Laboratories, Inc.: GARI-051101.

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
