# Peer review of "Neural representation of face familiarity in an awake chimpanzee"

_PeerJ, doi:10.7717/peerj.223_

## Round 0.1 · original submission · Minor Revisions

· Academic Editor

Minor Revisions

Dear Dr. Fukushima,
I am very pleased to inform you that the two reviewers examining your manuscript have expressed quite positive evaluations. They both suggest minor changes which I agree with.

Reviewer 1 ·

Basic reporting

General note: In macaque physiology it is a standard to use at least two if not three individuals, even if surgical and training procedure take an immense amount of time. I generally do not agree on these new standard of one subject studies and more critically on a series of reports in various papers/journals using one individual only. Hence, for chimpanzee ERP studies, I would hope that this type of case studies is not what will be the standard paradigm. I am fully aware of the effort it takes to train a chimpanzee, however, I am also aware of what it takes to train and prepare a macaque for an experiment. One talking pig is sufficient to prove that pigs can talk, goes a well-known argument in favor of single-case studies. However, we are not dealing with a talking pig here. For the sake of high-quality research, it would be nice to have more than one and to follow generally agreed standards of physiological animal testing.
Having this said, I clearly see that for the authors there is not much need to add additional subjects, since a couple of papers have been published as single subject studies. These papers however have a clear favor in terms of novelty despite the lack of validity due to low number of subjects.

Content:
Familiarity has been discussed and conceptually intermingled with expertise (different species' faces). It would be advisable to keep these terms apart and clearly define.

Experimental design

The experimental design is straight-forward and valid. However, the way of offline processing of the EEG data is questionable. It is not a common way to filter EEG data at .1 to 20 Hz. In contrast, for many, this sampling procedure is a criterion for rejection. Additionally I realized that in Hirata, S.et al. Brain response to affective pictures in the chimpanzee. Sci. Rep. 3, 1342; DOI:10.1038/srep01342 (2013) the sampling was up to 30 Hz, which is still too low, but different from here. I suggest the authors provide more insight into their findings by providing a different bandpass filtering, e.g. .1-100 Hz, notch, or .1-160 Hz, notch, plus filtering of averaged data at 30 Hz. There are many more reliable ways than what has been presented here. It is important to note that filtering produces different results.

Why is this important here? Looking at T5, T6 in figure 2 and 3, there is a double peak, that could be due to the way of filtering.
Generally, I would like to see a different filtering and a justification for the type of filter used here.

The placing of the electrodes: It has been argued in the introduction, lines 62ff, that a distributed network is involved in the processing of faces, especially emotional faces. Of course, deep structures like the Amygdala are out of question, however, parietal and frontal areas (off from the midline) are interesting areas of interests. This has not been done here, although it comes for free.

Please correct me if I am wrong: experiment 1 consisted of 1575 trials from which only 115+126+52. In other words, 4 out of 5 trials have been rejected? All due to the chimpanzee' lack of attention? Comparable number of experiment 2.

Validity of the findings

The fact that the study found a major difference in familiarity along the midline is rather surprising to me, lines 228ff, figure 2. Is this a very broad signal arising from both hemispheres? Then a more detailed location, as suggested above by placing electrodes more laterally on these sites, would totally make sense.

Additional comments

For this specific study, I think it is well conducted, however, it needs some justifications and partly re-analysis (different filtering) as described above.

For the general line of research on chimpanzees using EEG, I would strongly suggest to get more than one subjects ready.

·

Basic reporting

This study is interested in the processing of familiar and unfamiliar faces in chimpanzee for own and other species faces. The authors are recording the ERPs in one subject while watching either own species faces (exp1) or other species faces (exp2). They found a clear electrophysiological difference for familiar and unfamiliar faces only for own species.
I found this study very interesting but think that the authors should amend part of their introduction. They are stating that the comparative approach should “elucidate the physiological and evolutionary origin of human cognition”. This is s too strong statement, the comparative approach can clearly help to address this issue but may not solve the problem itself.
In their review of the literature, I believe that they should (if the word limit allows it) cite work by Stefan Schweinberger who has been studying the familiarity effect of face processing in human. The authors right now are focusing on the N170 which is not really involved in recognition. De Haan et al; (2002), Carmel, D., & Bentin, S. (2002) were interested in the human brain response to other species faces and should also be cited.
Finally, the authors are limiting their review to the Chimpanzee literature but they may want to cite a review from Tsao about the neural network involved in face processing in macaques which present also a lot of similarities with the human system.

From their review the authors are not making any clear prediction in term of ERPs. considering their work published in 2010, were they expecting differences between familiar and unfamiliar faces early?

In the discussion, the authors are mentioning the difference between own/other species and the role of experience. de Haan et al. should then be cited as they found that human do not process other species faces in the same way than human faces.

Experimental design

the design is very good

Validity of the findings

the statistics are adequate. I wonder if the authors should not run an ANOVA for exp2 to make both analysis similar.

Additional comments

A very nice article.

---

## Round 0.2 · accepted · Accept

· Academic Editor

Accept

I think you answered correctly to reviewers' observations. Still agreeing with them that a larger number of subjects would certainly improve the statistical validity of the study, I also think that this kind of experiments have the merit to open new frontiers.